# Memory-Efficient, Limb Position-Aware Hand Gesture Recognition using Hyperdimensional Computing

Author
Institution
City, State, Country

## ABSTRACT

Electromyogram (EMG) pattern recognition can be used to classify hand gestures and movements for human-machine interface and prosthetics applications, but it often faces reliability issues resulting from limb position change. One method to address this is dual-stage classification, in which the limb position is first determined using additional sensors to select between multiple position-specific gesture classifiers. While improving performance, this also increases model complexity and memory footprint, making a dual-stage classifier difficult to implement in a wearable device with limited resources. In this paper, we present sensor fusion of accelerometer and EMG signals using a hyperdimensional computing model to emulate dual-stage classification in a memory-efficient way. We demonstrate two methods of encoding accelerometer features to act as keys for retrieval of position-specific parameters from multiple models stored in superposition. Through validation on a dataset of 13 gestures in 8 limb positions, we obtain a classification accuracy of up to 93.34%, an improvement of 17.79% over using a model trained solely on EMG. We achieve this while only marginally increasing memory footprint over a single limb position model, requiring 8× less memory than a traditional dual-stage classification architecture.

## CCS CONCEPTS

• **Computing methodologies** → **Instance-based learning**; **Multi-task learning**.

## KEYWORDS

hyperdimensional computing, gesture recognition, multi-context classification

**ACM Reference Format:**
Author. YYYY. Memory-Efficient, Limb Position-Aware Hand Gesture Recognition using Hyperdimensional Computing. In *tinyML '21: tinyML Research Symposium, March 22–26, 2021, Online.* ACM, New York, NY, USA, 8 pages. https://doi.org/XX.XXXX/XXXXXXX.XXXXXXX

## 1 INTRODUCTION

Hand gestures can be classified using pattern recognition of muscle activity from electromyogram (EMG) signals for human-machine interface applications [16, 22, 24] and control of upper-limb prostheses [6, 9]. Various machine learning algorithms (including support vector machines (SVM) [25], linear discriminant analysis (LDA) [32], and artificial neural networks [11]) have been applied to this problem, and some recent works have focused on in-sensor implementation of gesture recognition for wearable devices [1, 17, 21, 26]. A major barrier to the practicality of these systems is dataset shift between the ideal conditions in which they are trained and the unknown conditions in which they are deployed [2]. Since EMG is typically measured non-invasively from the surface of the skin, it is subject to many types of signal variation occurring during everyday use [13]. One particular challenge that has often been observed is classification error due to changing limb positions. Beyond potentially causing electrodes to shift, varying limb positions can also have significant effects on the underlying muscle activity measured by EMG [10]. These variations can cause accuracy degradation if overlooked during initial training, and they often necessitate more complex classification models.

Recently, hyperdimensional (HD) computing has emerged as a paradigm for fast, robust, and energy-efficient classification of biosignals [28]. In HD computing, features are represented as extremely high-dimensional vectors, or hypervectors (HVs). This enables classification to be implemented as a simple nearest-neighbor search among class prototype HVs. Training and updating HD classification models involves building or augmenting these prototypes through superposition of training examples, and researchers have been able to fully embed this flow in a wearable EMG-based gesture recognition device [21]. As part of that study, two different limb position contexts were considered, and the model could be updated to classify hand gestures well in both positions. Still, there was a slight degradation of classification accuracy by 3-4% compared to models trained specifically for each limb position context, with greater degradation likely to occur for a larger number of positions interfering with each other in superposition. There is a need, then, to expand the capacity of HD classifiers for more contexts while still relying on superposition for memory efficiency.

Properties relating orthogonal vectors in HD spaces can be leveraged to achieve this. It has been shown that orthogonal, high-dimensional context vectors can be bound to neural network weights to access task-specific models stored in superposition with other models [5]. A similar result has also been demonstrated for HD classification, where orthogonal HVs representing different tasks were used as keys to access task-specific prototypes [3]. These strategies could be applicable to EMG-based gesture recognition, with gesture classification in each limb position treated as a separate task. This enables a dual-stage approach, in which limb position is first determined to choose the best position-specific parameters.

*tinyML '21, March 22–26, 2021, Online*
© YYYY Association for Computing Machinery.
ACM ISBN XXX-X-XXXX-XXXX-X/XX/XX…$DD.DD
https://doi.org/XX.XXXX/XXXXXXX.XXXXXXX

Using these ideas as a foundation, we present a sensor fusion-based HD classification architecture where accelerometer signals are encoded into limb position context HVs, allowing for superposition of prototypes from multiple limb positions with less interference between each other. With this method, we can effectively perform dual-stage classification while storing the position-specific parameters in superposition, rather than separately. This allows us to achieve higher classification accuracy without the drastic increase in memory footprint that normally comes from dual-stage or cascaded classifiers.

We refer to this method of binding context HVs as context-based orthogonalization, and we present an analysis of its expected classification margin improvement, along with two architectures for projecting accelerometer-based limb position information into context HVs. The first, which most closely resembles dual-stage classification, requires identification of discrete limb position contexts that are each represented by an orthogonal context HV. Context HVs are selected during training and inference based on limb position labels and limb position classification using accelerometer signals, respectively. The second architecture we present directly encodes accelerometer features into context HVs without the need to define discrete contexts. We test both architectures using a new dataset collected with a single subject wearing an EMG and accelerometer acquisition device (Figure 1). Our results show that sensor fusion for context-based orthogonalization can improve classification accuracy for 13 gestures across 8 limb position contexts from 75.58% to 93.37% without introducing significant implementation costs.

## 2 RELATED WORK

The most commonly explored way to improve gesture recognition in multiple contexts is to augment training routines with a wider range of contexts outside of perfect laboratory conditions. Works have used training sets consisting of multiple limb positions [10, 18, 27], as well as other types of variations including temporal variation [18] and contraction effort level [19]. Because this significantly increases initial training effort, it is beneficial to move to a continuous learning framework for gradually incorporating new contexts. One major advantage for methods based on HD computing is the capacity for computationally efficient model updates that can be performed on the fly. Training or updating only requires new HVs to be bundled with previously learned prototypes through superposition, which can be approximated in hardware for in-sensor learning using a bit-wise merge operation [21]. However, training set augmentation methods generally see diminishing returns, especially for larger numbers of distinct contexts [27]. A potential avenue for improvement in HD classifiers is prototype optimization [15, 23], but the iterative nature of these methods makes them impractical for in-sensor implementation.

Because features based solely on EMG can overlap and take on more complex distributions when considering multiple limb positions, other methods have leveraged new input features or signal modalities that vary with and provide information about the limb position. Some works append accelerometer features to existing EMG feature vectors [10, 30], while others take a dual-stage classification approach [10, 12]. A separate gesture classifier can be trained for each limb position, with the most appropriate one

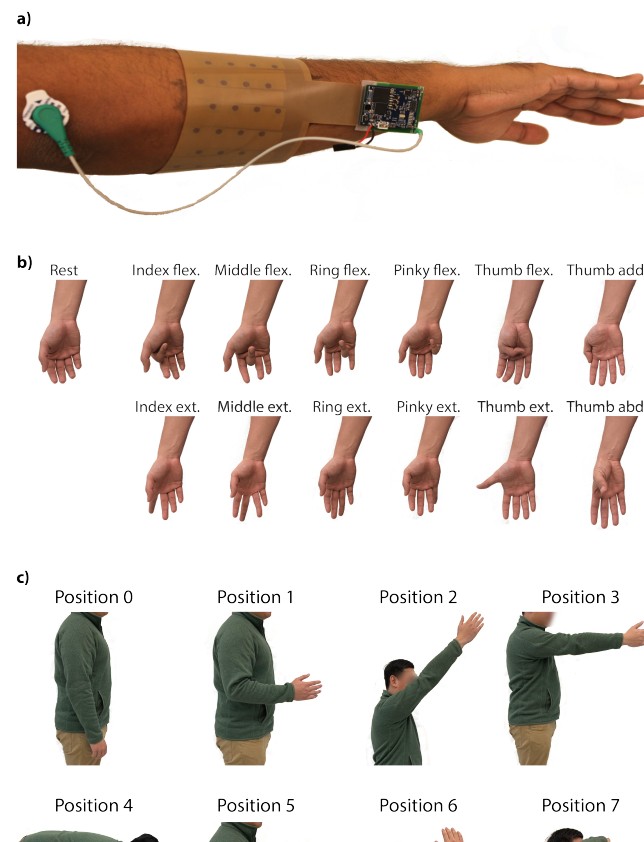

**Figure 1: Wireless EMG + accelerometer acquisition device with flexible printed circuit board electrode array (a). The dataset consists of 13 hand gesture classes (b) for classification in 8 limb position contexts (c).**

selected during deployment based on limb position classification using accelerometer signals and either an LDA [10, 12] or SVM [29] model. A hybrid dual-stage classification approach has also been suggested, which consists of multiple classifiers each trained in a subset of limb positions, using an accelerometer for coarser limb position detection [27]. While these architectures improve classification accuracy, they come with significant implementation overhead. Memory allocation for storage of model parameters increases linearly with the number of limb positions, and an additional limb position classifier must also be implemented. These drawbacks have made dual-stage classification unattractive for in-sensor implementation.

This paper addresses the problem of memory footprint for a dual-stage HD classifier by leveraging sensor fusion for parameter superposition. Sensor fusion for HD classifiers has previously only been presented in a single-stage framework [4]. In that work, HD computing was used to combine three classifiers, each operating on

a different feature modality but performing the same classification task in a single context. This is in contrast to the strategy proposed in this work, where we approximate the use of different sensor modalities for each stage of dual-stage classification for multiple contexts. Our context-based orthogonalization method is based on work done using orthogonal context vectors to minimize interference between different model parameters stored in superposition [3, 5]. Here, we further analyze the benefits of this method when applied to HD computing, and we demonstrate ways to incorporate sensor fusion for creating context-specific HVs using both context classification and direct encoding of accelerometer signals.

## 3 DATA ACQUISITION

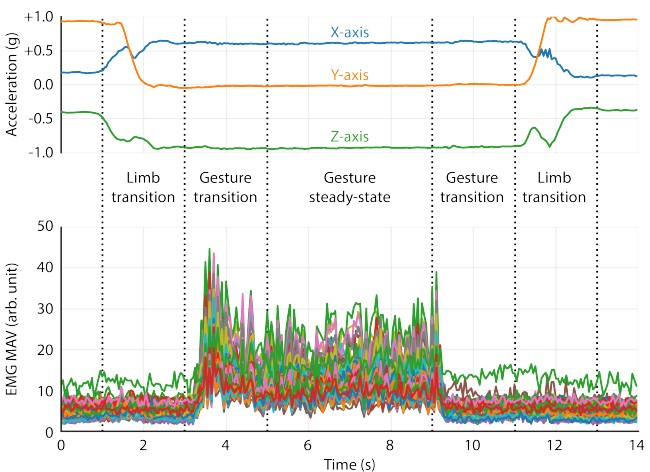

**Figure 2: 64-channel EMG mean absolute value (MAV) features and 3-axis accelerometer features from an example repetition of middle finger extension in limb position 3, with annotated timing directions.**

To test our context-based orthogonalization methodology, we collected a new dataset with simultaneous EMG and accelerometer measurements during the performance of multiple gestures in multiple limb positions. Data were acquired using a custom wearable EMG device enabling compact and wireless biosignal acquisition, processing, and streaming [21]. For this study we utilized a 64-channel electrode array implemented on a flexible printed circuit board, and we added an on-board accelerometer (InvenSense MPU-6050) to measure limb position. The electrode array was wrapped around the largest section of the dominant forearm to cover the extrinsic flexor and extensor muscles involved in digit movements. The device was oriented such that the accelerometer was positioned over the ulna approximately two inches from the wrist, measuring the movements of the forearm. For this study, the device's in-sensor learning capabilities were not used, and raw EMG and accelerometer signals were wirelessly streamed to a laptop for offline analysis. Figure 1 shows the gesture classes and limb positions used for the study. The 13 gesture classes included 12 single degree-of-freedom movements of each digit along with the rest class (no movement). Eight different limb positions were chosen based on literature to mimic ones that may be encountered in everyday use [29].

The experimental procedure consisted of one recording session per limb position, in which each gesture was performed for three repetitions. A single repetition (Figure 2) consisted of beginning in the default limb position (Position 0) and the rest gesture, transitioning to the directed limb position within a two-second window, performing the directed gesture within a two-second window, holding the gesture for four seconds, transitioning back to the rest gesture within a two-second window, and returning to the default limb position within a two-second window. A relaxation period of three seconds followed each repetition. Directions for gesture and limb position and timing guidance were provided to the subject through a custom graphical user interface, which also logged the wirelessly streamed data. For analysis, only the central four-second steady-state period of the gesture performance was used. Each period was divided into 50 ms windows for feature extraction of the average x-, y-, and z-axis acceleration and mean absolute value of each of the 64 EMG channels.

## 4 CONTEXT-AWARE HD COMPUTING

In this section, we describe our context-based orthogonalization method for gesture classification in multiple limb position contexts. We first show how EMG features are projected into HVs and classified using a baseline HD classifier with direct superposition (Figure 3a). Next, we show how a dual-stage architecture (Figure 3b) is emulated using context-based orthogonalization (Figure 3c), and we analyze the expected classification margin improvement. Finally, we demonstrate how accelerometer features can be directly encoded into context HVs (Figure 3d).

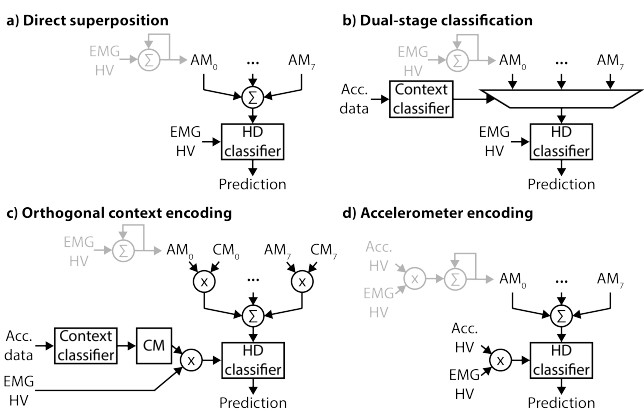

**Figure 3: Classifier architectures for multiple limb position contexts, with training process in gray. a) Direct superposition of limb position-specific prototypes without context information. b) Dual-stage classification with limb position classification to select position-specific prototypes. c) Context-based orthogonalization using classified limb position to enable superposition of prototypes. d) Direct encoding of accelerometer (Acc.) signals as context vectors without limb position classification.**

## 4.1 EMG Data Projection

HD computing for classification tasks involves projection of extracted features into HVs of extremely high dimensionality (e.g. D = 10,000) [28]. Key building blocks of HD projection architectures include the random item memory (IM) for representation of categorical or symbolic information, a continuous item memory (CIM) for representing ranges of quantized values, and algebraic operations to manipulate and combine HVs from these memories. These operations include element-wise addition $(+)$ and multiplication $(*)$ between HVs, permutation $(\rho)$ of the elements within an HV, and scalar multiplication between an HV and a scalar value. For bipolar $(\{-1, +1\}^D)$ HVs, addition is implemented as element-wise majority with random tiebreaks and is used for superposition of multiple HVs during training. Element-wise multiplication can be seen as binding one HV to another, and binding with a randomly generated HV has the effect of rotation into an orthogonal part of the HD space. This property is central to context-based orthogonalization.

Projection operations are designed to encode various relationships between inputs, and we implement an architecture that has been successfully used for encoding and classifying EMG features [20, 21]. We represent each electrode channel using an orthogonal 10,000-D bipolar HV stored in an IM. First, spatial encoding is performed by computing a bipolarized weighted sum of these HVs using the extracted feature values from each channel as weights. Temporal information is then encoded using a permute and multiply operation across five consecutive spatially encoded HVs. Thus, each encoded EMG HV represents 250 ms of EMG data, with an overlap of 200 ms.

## 4.2 Classification

Classification of projected query EMG HVs is performed through nearest neighbor search among elements of an associative memory (AM), in which prototype HVs representing the different classes are stored. Class prototypes are calculated by superimposing all training examples from a particular class through element-wise majority, and similarity between prototypes and a query HV is measured using Hamming distance. A minimum of one prototype per output class is required, although it is also possible to store multiple different prototypes representing same class [19]. We would like to minimize the size of this AM to reduce both memory footprint and cycles of operation for similarity metric calculations. To do this, all training HVs for a particular gesture, regardless of limb position, should be directly superimposed into a single prototype.

The source of error due to direct superposition of training examples from multiple contexts (Figure 3a) can be analyzed if we replace the element-wise majority operation with integer addition, and if we replace Hamming similarity between two HVs with the dot product $(\cdot)$. The entry in the AM representing gesture $g$ can be calculated as $AM[g] = \sum_p AM_p[g]$, where $AM_p[g]$ is the position-specific prototype for gesture $g$ in position $p$. Classifying a query HV $X$ with gesture label $y$ from limb position context $c$ is then computed as:

$$\hat{y} = \underset{g}{\operatorname{argmax}} \left( X \cdot AM[g] \right)$$
$$= \underset{g}{\operatorname{argmax}} \left( X \cdot AM_c[g] + \sum_{p \neq c} X \cdot AM_p[g] \right) \quad (1)$$

We split each dot product into a position-specific classification term and an error term from interference with other contexts. This error depends on similarity between prototypes from different gestures and different positions. If we assume that position-specific classification is accurate, then the goal for context encoding is to minimize interference error by orthogonalizing prototypes from different limb positions.

## 4.3 Context Encoding

*4.3.1 Context-Based Orthogonalization.* Context-aware classification can be achieved using a dual-stage architecture as depicted in Figure 3b, which resolves the issue of interference between contexts by separating them completely. However, this method would require separate storage of context-specific parameters and scale memory footprint by the number of desired contexts. We can instead orthogonalize gesture prototypes from different limb position contexts by binding them with orthogonal context vectors prior to superposition [3, 5]. During inference, query hypervectors are also bound to their corresponding context vectors before nearest neighbor search.

Here, we show that context-based orthogonalization effectively enables dual-stage classification while still allowing for superposition of parameters and thus memory efficiency. Figure 3c depicts the implementation for an HD classifier. Entries in the AM are now calculated as $AM^*[g] = \sum_p CM_p * AM_p[g]$, where $CM_p$ is a context memory HV representing limb position $p$ and is orthogonal to context HVs representing other positions. Classifying a query HV is now computed as:

$$\hat{y} = \underset{g}{\operatorname{argmax}} \left( CM_c * X \cdot AM^*[g] \right)$$
$$= \underset{g}{\operatorname{argmax}} \left( CM_c * CM_c * X \cdot AM_c[g] \quad + \right.$$
$$\left. \sum_{p \neq c} CM_c * CM_p * X \cdot AM_p[g] \right) \quad (2)$$
$$= \underset{g}{\operatorname{argmax}} \left( X \cdot AM_c[g] + \epsilon \right)$$

Because binding an HV with itself results in an identity vector, the first term of the dot product is again a position-specific term. Binding two random HVs with each other results in a new, random HV, diminishing the dot product error term.

*4.3.2 Classification Margin Analysis.* A classification margin can be defined by viewing the task as a problem of noisy element retrieval from the AM, with error due to both sample-by-sample variation and prototype superposition. Since sample-by-sample variation should be zero-mean, we neglect it in defining retrieval error for class $y$ in context $c$:

$$d_{retrieve} = d \left( CM_c * AM_c[y], \sum_p CM_p * AM_p[y] \right) \quad (3)$$

where $d(a, b)$ is the Hamming distance function. We can compare this with the distance to the nearest incorrect class:

$$d_{wrong} = d \left( CM_c * AM_c[y], \sum_p CM_p * AM_p[g \neq y] \right) \quad (4)$$

to approximate the classification margin:

$$m = \left( d_{wrong} - d_{retrieve} \right) / \left( d_{wrong} + d_{retrieve} \right) \tag{5}$$

For the baseline case of direct superposition, we can just set all of the $CM_p$ context vectors to be identical. We simulate a general classification task using randomly generated prototypes with average pairwise distance of $d_0$, regardless of class or context. Figure 4a shows the retrieval error and incorrect classification distance when superimposing 11 contexts with varying $d_0$, both for direct superposition and with context encoding. While orthogonal context encoding increases retrieval error to a constant value, it also increases incorrect classification distances by a larger amount, leading to a larger classification margin as shown in Figure 4b. This improvement in margin is shown for varying odd numbers of contexts in Figure 4c. There is no improvement in margin when there are three or fewer contexts, or in extreme cases where all prototypes are identical or orthogonal. Otherwise, random context encoding always improves margin, with the maximum improvement occurring with 11 contexts and $d_0 \approx 0.3$.

*4.3.3   Sensor Fusion of Accelerometer Signals.* We explore two different ways to generate context vectors using accelerometer features. First, we can perform dual-stage classification and train a first stage classifier on accelerometer signals to select between limb position context HVs. We can classify limb positions using a simple linear SVM, and a context memory can be created with a randomly generated, orthogonal vector representing each limb position. Figure 5a shows the accelerometer features for different limb positions used to select between context memory hypervectors.

Alternatively, we can directly encode accelerometer signals into context HVs, without the need to classify limb position (Figure 3d). Accelerometer signals are first quantized between $+1g$ and $-1g$ into discrete levels which can be represented by HVs from a CIM (Figure 5b). A separate memory is used for each of three accelerometer axes, and encoded quantization levels are then bound together to form a single context HV. The CIM preserves relationships between quantization levels, representing extreme values ($\pm 1g$) with two HVs $d_{max}$ Hamming distance apart and representing intermediate values as a mixture of these HVs. This ensures similarity between examples from the same limb position, while still differentiating between HVs representing different positions.

## 5   EXPERIMENTS

### 5.1   Dual-Stage Classification and Direct Superposition

To provide baseline comparisons with our context-based orthogonalization methods, we implemented two strategies from prior art for multi-limb position classifiers. The first was a true dual-stage classifier (Figure 3b), where we used an SVM to classify accelerometer signals and choose an EMG AM containing position-specific prototypes. The second was direct superposition of each of the limb position-specific AMs without any context information (Figure 3a). Classification accuracy for all experiments was calculated using 10-fold cross-validation of the dataset.

The first two columns in Figure 6 show the results for these baseline strategies. Dual-stage classification, as expected, achieved the best-case scenario classification accuracy of 98.46% across all

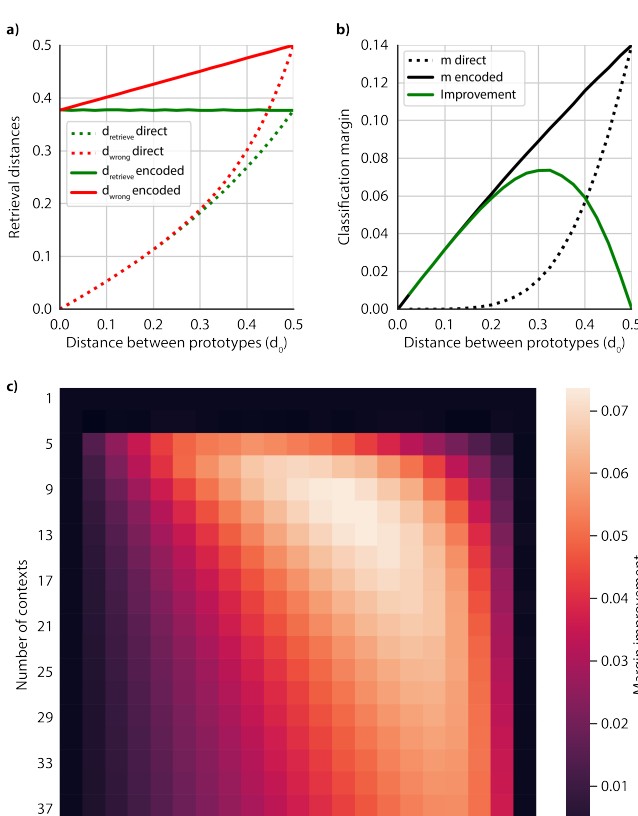

**Figure 4: Improvement in classification margin using orthogonal context encoding. a) Retrieval error (green) and incorrect classification distance (red) for 11 contexts. Dashed lines are with direct superposition, and solid lines are with context-based orthogonalization. b) Classification margins and margin improvement due to orthogonal context encoding for 11 contexts. c) Margin improvement as a function of both the number of contexts and distance between classes.**

limb positions. However, this method has the major drawback of requiring separate EMG and limb position classification modules, as well as $> 8\times$ the required parameter storage compared to a single limb position model. On the other hand, the direct superposition model is identical to a limb position-specific model in terms of implementation, but it results in an accuracy degradation of 22.88%.

### 5.2   Dual-Stage Context Based Orthogonalization

Orthogonal context encoding using a limb position classifier (Figure 3c) provided the most dramatic improvement in classification accuracy to 93.37%, while only increasing memory footprint by 3.7% compared to the direct superposition case and limb position-specific models (Figure 6). The marginal increase in parameter memory was for storage of SVM parameters. For our tests, 99.99% accuracy in

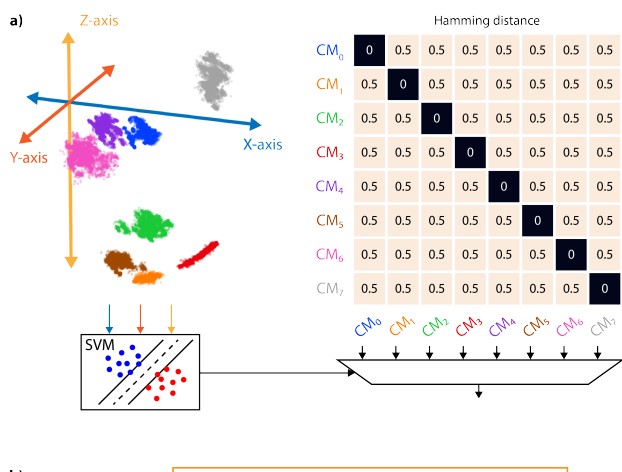

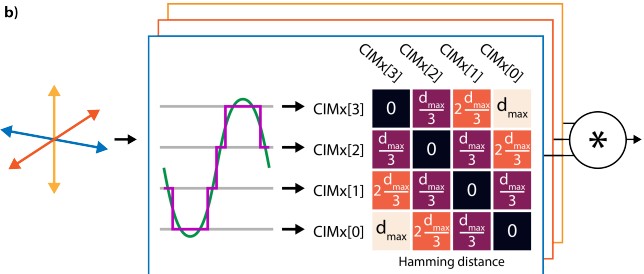

**Figure 5: Different methods for projecting accelerometer signals into hypervectors. a) Selection of a context vector from an orthogonal context memory based on classified limb position contexts using SVM. b) Using a continuous item memory to represent the quantized signal from each accelerometer axis and binding the result.**

limb position classification could be achieved while using only up to 50 stored support vectors consisting of 32-bit floating point accelerometer features. Including the HD classifier prototypes, which require 10,000 bits for each of the 13 classes, the total parameter memory footprint is 134.8 kb.

Although light weight in terms of memory requirement, the use of a limb position SVM does still require implementation of the decision function, adding to the model complexity. Additionally, the SVM must be trained using discrete, labeled limb positions decided upon before deployment. This limitation motivates direct encoding of accelerometer signals to create an arbitrary number of context vectors.

### 5.3 Accelerometer-Based Context Encoding

We used a genetic algorithm [31] to determine optimal parameters for encoding accelerometer features into context vectors (Figure 3d). These parameters were chosen separately for each accelerometer axis, and they included the number of quantization levels as well as the distance $d_{max}$ between extreme values in the CIM. Optimal parameters are listed in Table 1.

CIM-based accelerometer encoding improved classification accuracy by 13.61% averaged over all positions while removing the

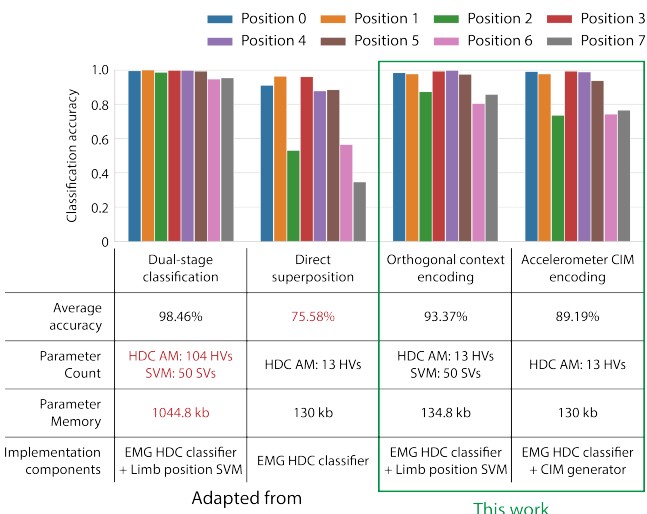

**Figure 6: Comparison of different context encoding methods, including average accuracy over all 8 limb positions, number of parameters including EMG hypervectors (HVs) and limb position support vectors (SVs), memory allocation for parameters, and required high-level components for implementation.**

**Table 1: Optimal CIM encoding parameters for accelerometer signals.**

| Parameter | Value |
|---|---|
| X quantization levels | 59 |
| X $d_{max}$ | 1.0 |
| Y quantization levels | 55 |
| Y $d_{max}$ | 0.5 |
| Z quantization levels | 14 |
| Z $d_{max}$ | 0.6 |

required overhead for a limb position classifier (Figure 6). Generation of CIM HVs can be achieved through re-use of components already existing in the main EMG projection architecture, adding very little to implementation complexity. A slight decrease in performance compared to orthogonal context encoding is expected, as context HVs from different limb positions can no longer be guaranteed to be orthogonal (Figure 7). Because certain pairs of limb positions are more similar to one another (e.g., positions 0 and 4), those limb positions are weighted more heavily in the superposition and can hurt performance in the remaining positions. Still, performance was improved for all limb position contexts as compared to direct superposition, while removing the requirement for limb position classification.

The uneven performance across the different limb position contexts could potentially be addressed by optimizing the placement of the accelerometer or adding additional sensor modalities to maximize feature variation across limb positions and generate more

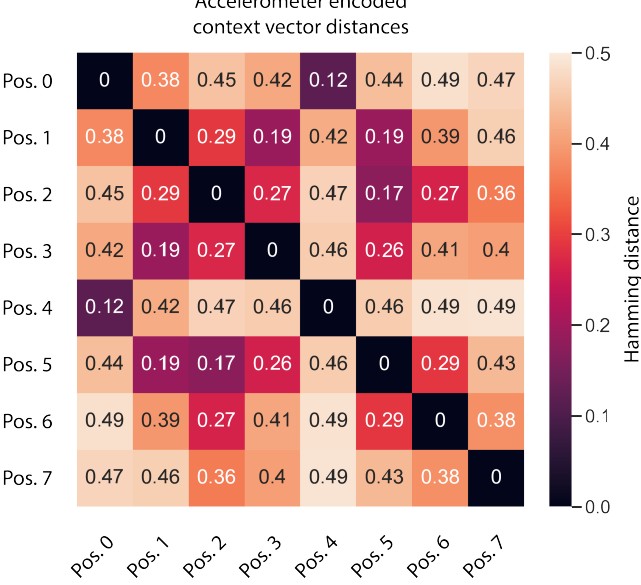

**Figure 7: Distances between limb position context vectors formed using accelerometer signals encoded with a continuous item memory.**

distant context HVs. This would reduce the number of limb positions with overlapping context features, mitigating the issue of context weighting in superposition. An alternative way to address this would be to prune training examples and ensure certain contexts are not weighted more than others. For this strategy, similar limb positions like positions 0 and 4 would be considered as a single context in superposition, and only half of the training examples from each position should be used.

## 6  CONCLUSION

In this work we presented a sensor fusion-based method for context-aware classification of EMG signals for gesture recognition. By using accelerometer signals to bind context information to existing EMG parameters, we could achieve high classification accuracy in 8 limb positions with a model that does not increase memory requirement over a single limb position. We first evaluated the improvement in classification margin when using context-based orthogonalization for an HD classifier, and we highlighted its dependency on relationships between class prototypes and the number of contexts to be learned. We proposed two methods to generate context vectors for different limb positions using accelerometer signals, with one providing better accuracy when distinct limb positions were predetermined, and one allowing for a continuous range of limb positions. Both methods require minimal additions to a baseline HD computing gesture classifier, which can already be fully implemented on a wearable device [21]. Our results point towards a way to improve reliability of gesture recognition for wearables in everyday situations, and our method can potentially be applied to other multi-context biosignal classification applications [7, 8, 14].

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
