# OpenReview forum: "Memory-Efficient, Limb Position-Aware Hand Gesture Recognition using Hyperdimensional Computing"
_tinyml.org/tinyML/2021/Research_Symposium — tinyML 2021 Regular_

### Official Review · AnonReviewer4 · 2021-01-27

**Overall Merit Score:** 3

**Brief Summary:**

The paper presents a model for hand-gesture recognition. The model is based on hyperdimensional computing. The key idea is to include the limb position context for hand-gesture recognition via sensor fusion. The inclusion of the context improves accuracy from 75 to 93%. It also evaluates parameter memory which is about 130 kb.



**Detailed Comments:**

The work aims to improve accuracy and address it with the sounding approach of including the context information. The resulted model seems to improve the baseline substantially. The paper misses several key pieces of information in the comparison to the state of the arts.

**Paper Strengths:**

The proposed approach of context inclusion seems to be effective in terms of accuracy.


**Paper Weaknesses:**

It is reasonable but not ground-breaking work.
The hardware cost analysis has room to improve. The actual mapping of the model on a microcontroller would strengthen the paper.
The comparison table should indicate/refer to the prior works.

**Poster (If Paper Is Rejected):**

1: Yes, ok for poster sesion to nurture work

**Reviewer Confidence:**

3: The reviewer is fairly confident that the evaluation is correct

---

### Official Review · AnonReviewer3 · 2021-01-30

**Overall Merit Score:** 3

**Brief Summary:**

The authors proposed to append the accelerometer value, which is able to track limb position, for predicting hand gesture via hyperdimensional computing. This is because the performance of Inferencing hand gesture using EMG changes a lot depending on limb position. Two architectures of binding accelerometer information were proposed: 1) a context hypervector corresponding limb position classified by accelerometer using SVM is bound with an EMG hypervector 2) a context hypervector directly encoded from accelerometer is bound with an EMG hypervector.  They showed reduced accuracy compared to HDC dual-stage baseline, which uses different HDC parameters for different limb position. However, they showed small memory footprint for storing parameters, because one parameter set is adopted to all the limb position.

**Detailed Comments:**

Measurement result is attractive. Adopting HD for the specific hand gesture application showed good accuracy result. However, it is not enough compelling mainly because of lack of comparison. The authors compared the novel work only with the HDC baseline. Same accelerometer sensor fusion work has been done using LDA & SVM for same application (referenced in paper by [10], and [30]). In terms of memory efficiency, the referenced ones would be better. Additionally, there is no detailed comment about hardware component cost and the number of operations. The author only compares the memory storage quantitively.

Consequently, this paper is acceptable if we considered this as an embedded classification system among HDC classification system. Otherwise, there seems to be room to improve the paper by comparing with other algorithms having the same sensor-fusion concept.

**Paper Strengths:**

-	Adopted hyperdimensional computing on hand gesture recognition
-	Good accuracy result with small memory footprint


**Paper Weaknesses:**

-	Lack of comparison with references having same concept of sensor fusion on the same application
-	Only memory footprint was emphasized
-	Lack of analysis of HW cost and the number of operations


**Poster (If Paper Is Rejected):**

1: Yes, ok for poster sesion to nurture work

**Reviewer Confidence:**

4: The reviewer is confident but not absolutely certain that the evaluation is correct

---

### Official Review · AnonReviewer2 · 2021-01-31

**Overall Merit Score:** 3

**Brief Summary:**

This paper presents a method to classify gestures using a combination of EMG and accelerometer data using hyperdimensional computing.

**Detailed Comments:**

- This paper presents a method to classify gestures using a combination of EMG and accelerometer data using hyperdimensional computing.

- Achieves high accuracy on the task with small increase in memory requirement.

- The paper requires the reader to have familiarity with HD computing.

- There is not much context given for how well non HD methods perform on this classification task.

**Paper Strengths:**

- Achieves high accuracy on the task with small increase in memory requirement.

**Paper Weaknesses:**

- The paper requires the reader to have familiarity with HD computing.

- There is not much context given for how well non HD methods perform on this classification task.

**Poster (If Paper Is Rejected):**

1: Yes, ok for poster sesion to nurture work

**Reviewer Confidence:**

1: The reviewer's evaluation is an educated guess

---

### Decision · Program_Chairs · 2021-02-05

**Decision:**

Accept (Regular)

**Comment:**

Congratulations on your paper's acceptance!

Your paper has been accepted as a full-length regular paper.

Please read the reviews carefully and make sure the concerns are addressed in your final submission.

All accepted papers will be given a slot in the TinyML Summit schedule for an oral presentation on Friday, March 26, 2021.

Camera ready instructions will follow soon. All papers will be hosted on arXiv and published papers will have the following header stamp: “Published as a conference paper at TinyML Research Symposium 2021.” The paper will also be presented on the program website.